# Emergency department is not safe anymore: Nurses describing their suffering

Ahlam Al-Natour*, Lubna Abuziad

Community and Mental Health Department, School of Nursing, Jordan University of Science and Technology, Irbid, Jordan

* asalnatour@just.edu.jo

## Abstract

### Background

Workplace violence represents a critical and alarming crisis in healthcare settings around the globe. This study seeks to shed light on the experiences of Jordanian nurses working in emergency departments, focusing on the forms of workplace violence they encounter, their emotional responses, and their coping strategies.

### Methods

Utilizing a qualitative descriptive design, we engaged a purposive sample of 24 nurses from two public hospital emergency departments. Data were gathered through in-depth, semi-structured interviews, and analyzed using the seven-step Colaizzi method. Four main powerful themes emerged from the interviews:: (1) the shocking and harsh experience of nurses with WPV, (2) the negative consequences of WPV, (3) nurses feelings toward their experience of WPV, and (4) coping strategies for dealing with WPV.

### Conclusion

The ramifications of workplace violence on nurses' physical and mental health are profound, significantly influencing their intentions to remain in their positions. It is imperative for nursing policymakers and hospital leaders to take decisive action against workplace violence. Effective anti-workplace violence policies must be rigorously enforced by hospital authorities to safeguard the health and well-being of emergency nurses, thereby fostering a safer and more supportive healthcare environment for all.

**Data availability statement:** All relevant data are within the manuscript and its Supporting Information files.

**Funding:** The author(s) received no specific funding for this work.

**Competing interests:** The authors have declared that no competing interests exist.

## Introduction

Workplace violence (WPV) is an alarming global and significant phenomenon in all healthcare settings [1]. It is considered one of the most essential epidemics and social occupational problems [2]. WPV includes a broad spectrum of unacceptable behaviors with staff being abused, threatened, discriminated, or assaulted in their work, which is a threat to their safety, health, and well-being [3].

Forms of WPV include physical violence, verbal abuse, and sexual abuse. Physical violence includes beating, kicking, slapping, serious injuries, or death against another person or group [4]. Verbal abuse includes acts of devaluing, humiliating, using offensive language, yelling, or screaming with the intent to offend and frighten a person via written or spoken word at work [4,5]. Sexual abuse includes unwelcomed or uninvited actions that involve physical contact, sexual acts, and harassment [5].

Nurses who work in the Emergency Department (ED) are some of the most vulnerable health workers to WPV due to their work on the frontline. So, they are vulnerable to different forms of WPV committed by patient or their relatives. A descriptive cross-sectional study about WPV in the ED in Taiwan revealed that 91.2% of nurses reported verbal abuse and 75.2% reported physical abuse by patients [6].

In recent years, WPV increased in Jordanian hospitals, especially in the ED, and occurred at high prevalent rates. A study conducted among nurses in rural public hospitals in Jordan revealed that 49.1% and 55.5% of nurses experienced physical and verbal violence [7].

Another study conducted at the ED in Jordan revealed that 33% of nurses and physicians reported physical violence and about 53% verbal violence by patients and their relatives while working [8]. WPV ends with negative consequences on nurses' health including increased emotional distress and inflexibility, feelings of fear, irritability, anger, restlessness, sleep problems, frustration, loss of interest, exhaustion, absenteeism, and injuries or even death [9]. The effects of WPV were increased work errors, nurses' burnout, turnover intentions, and decreased levels of productivity and job satisfaction [10].

Emergency nurses have observed that incidents of workplace violence (WPV) are significantly underestimated and often go unreported. Research studies underscores the urgent need for targeted educational interventions and training programs focused on WPV prevention. By equipping nurses with the knowledge and tools necessary to recognize and address these incidents, we can empower them to report WPV occurrences promptly and effectively [11,12].

It is crucial to explore this phenomenon in greater depth, as a thorough understanding can inform the development of impactful strategies to combat this pressing issue in emergency departments (EDs). Currently, there is a notable scarcity of descriptive qualitative studies that investigate the full range of experiences emergency nurses encounter with WPV. This study aims to provide a detailed account of these experiences, identifying the most common forms of WPV faced in the ED. Furthermore, it seeks to delve into the emotional responses of Jordanian nurses toward

WPV and the coping mechanisms they adopt in the aftermath of such incidents. By shedding light on these critical issues, we can drive meaningful change within the healthcare system.

## Method

### Study design and setting

A descriptive qualitative design was utilized for this study. This approach enhances the understanding of the underlying experiences and perceptions related to the phenomenon being studied, as experienced by the participants in its natural context [13]. The research study was conducted at two hospitals in a Northern city in Jordan, specifically a university hospital (X) and a governmental hospital ED. Data collection took place in a private room within the ED. These selected hospitals are the largest tertiary public hospitals in the city, providing care for the majority of residents in Jordan.

### Study participants

A purposive sampling method was utilized for this study to provide a detailed description of the experiences of nurses who have encountered workplace violence (WPV). This type of sample specifically focuses on participants who have experienced the phenomena being studied [14]. The research involved 24 registered nurses—12 males and 12 females—currently employed in the emergency department (ED) of two hospitals, with 12 participants from each hospital.

This descriptive study aimed to conduct an in-depth exploration of narratives related to the phenomenon under study. Thus, 24 participants were deemed an adequate sample size to achieve saturation of information. The inclusion criteria were as follows: participants had to be Jordanian nurses aged between 23 and 55 years, possess a bachelor's degree, be fluent in Arabic, have worked in the ED for over one year, and have experienced WPV at least once.

### Data collection process

Before data collection began, a letter outlining the nature and significance of the study was sent to the administrators of two hospital to obtain their approval and support. Two researchers conducted face-to-face interviews with the study participants. The primary investigator (PI) was an expert in qualitative research while the second researcher had received training in the principles of qualitative studies and data collection through interviews. The PI complied a list of nurses working in the EDs and scheduled interviews at time and location that were convenient for the participants.

The study researchers personally contacted each participant and invited them to participate in the study. Participants verbally agreed to participate and share their experiences of WPV. Before the interview, each participant signed a written consent form. The interviews took place in a private room, and data were collected using an in-depth semi-structured approach

Each participant responded to a series of main open-ended questions along with additional probing questions. These questions were developed by the primary investigator (PI) and reviewed by an expert qualitative research. The questions were designed to align with the study's purpose, which aimed to capture the experiences of nurses dealing with workplace violence (WPV) in the emergency department (ED). The specific questions asked were:

1. Describe your experience with WPV in the ED.

2. How many times have you encountered WPV? Please describe a specific incident.

3. What forms of WPV have you experienced in the ED?

4. Describe your feelings immediately after the incidence of WPV and afterward.

5. What are the consequences of WPV on your physical and mental health?

6. How does WPV affect your quality of work, job satisfaction, and likelihood of turnover?

7. Describe the impact of WPV on your health institution.

8. Do you feel safe in your work environment? Why or why not?

9. How did you cope following the incident of WPV?

Interviews were audiotaped and field notes were taken. Each interview lasted between 30–45 minutes. Data collection continued utile saturation was reached, meaning no new themes emerged.

## Ethical considerations

Before data collection, institutional research board (IRB) approval was obtained from the university (x) hospital and the Ministry of Health (IRB: 147/132/2020). Additionally, administrative permission was granted by the selected hospitals for conducting the study. Participants were provided with a thorough explanation of the study's purpose, procedures, risks, and benefits. All participants signed consent forms to confirm their voluntary participation and were informed of their right to withdraw from the study at any time. All participants also approved to have their interviews audio-recorded. Interviews with participants took place in private rooms at the EDs of the two hospitals. The data were transcribed verbatim, and each hard copy of the transcripts was assigned a serial number to ensure confidentiality. All hard copies were stored securely in a locked cabinet belonging to the primary investigator.

## Data analysis

Collected data were analyzed using the 7 steps of Colaizzi [15]. The analysis process is presented in Table 1.

## Trustworthiness

Data trustworthiness was established and verified based on the following key criteria: credibility, dependability, and transferability [16]. Credibility was ensured through triangulation and member checking approaches. Dependability was confirmed by conducting data audit and and providing detailed description of the study phenomenon. Transferability was achieved through purposive sampling which ensure that data collected was specific to the study context.

Table 1. Steps of Colaizzi (1978) of data analysis.

|  | Coliazzi step | How steps implemented in the analysis process |
|---|---|---|
| 1. | Read and re-read the participant's transcripts to make sense of them | - Participants' transcripts were reviewed several times by two researchers to understand the description of WPV perception of study participant's experiences. This helped me understand and be familiar with the study phenomenon. |
| 2. | Extract significant statements from the transcripts | - Significant phrases and statements that directly involved and described the study phenomenon were extracted |
| 3. | The process of giving the meaning to the extracted statements | - The meanings of these phrases and extracted sentences were formulated. |
| 4. | Generate main study themes | - Main themes and subthemes are extracted and categorized into groups based on the formulated meaning |
| 5. | Integrated themes were defined in a comprehensive description of the results | - Each theme and sub-theme compile a detailed description of the study phenomenon |
| 6. | The basic structure of the description of the study phenomenon discussed | - The basic structure of the description of nurse's perceptions and experiences of WPV was identified. |
| 7. | Validate study findings | - Findings of the study were validated by expert reviewers and study participants to avoid researcher bias and prejudgment during and after data analysis. |

**Table 2. Study participant's characteristics (N = 24).**

| Variable | n (%) | Mean (range) |
|---|---|---|
| 1. Age | 24 (100%) | 34.6 (27–46) |
| 2. Sex | | |
| Male | 12 (50%) | |
| Female | 12 (50%) | |
| 3. Experience at ED | 24 (100%) | (1–5 years) |
| 4. Marital status | | |
| Married | 18 (75%) | |
| Not married | 6 (25%) | |
| 5. Hospital | | |
| Princess Basma | 12 (50%) | |
| University Hospital | 12 (50%) | |
| 6. Monthly income (JD*) | 24 (100%) | 500–700 JD |

JD: Jordanian Dinar

## Results

Twenty-four Jordanian nurses working in the ED at two major hospitals in Jordan participated in this study. Their ages ranged from 27 years to 46 years. All the participants were registered nurses with bachelor's degree, and had 1–5 years of experience in the ED. Data summarizing the participants' characteristics are presented in Table 2.

Four major themes emerged from the analysis of the participants' verbatim transcripts. These themes were: (1) the shocking and harsh experience of nurses with WPV, (2) the negative consequences of WPV, (3) nurses feelings toward their experience of WPV, and (4) coping strategies for dealing with WPV.

### Theme 1: The shocking and harsh experience of nurses with WPV

Participants described their experiences with workplace violence (WPV) as hazardous, unusual, and unexpected. They found these experiences particularly challenging and shocking, especially at the beginning of their careers as emergency nurses. Nurses expressed surprise at the verbal and physical aggression they faced from patients while providing care. Nurse 20 said,

> "The first incident of WPV was a harsh experience. It was the most devastating and unforgettable one. I was performing an ECG on an elderly woman, and her husband insisted on entering the room with her. The room was very small, so I politely asked him to wait outside and closed the door. After five minutes, when I finished, I opened the door, and he started shouting, saying, 'You need to be more polite and respectful; I want to teach you this!' He then tried to raise his hand to hit me. Thankfully, the medical staff and a policeman prevented him from doing so. I wondered: I was just doing my job; why are we treated this way?"

Participants reported that patients who were seeking medical care in the ED, or their relatives were often the perpetrators of WPV. Nurses indicated that WPV occurred in a high frequency particularly at night, and primarily targeted female nurses, with incidents happening on a daily basis. Despite this, the nurses in the study noted that the incidents of WPV were rarely reported by victimized nurses to hospital authorities. Nurse 19 stated,

> "Certainly, evevery day and every shift we experience WPV, which seems to be increasing with time. However, most WPV incidents are managed without being reported to hospital authorities. These incidents stem from relatively simple

esituation. Patients' relatives are often under significant pressure; having brought the patient with difficult and emergency circumstances."

The study participants reported several forms of WPV including verbal, physical, and sexual violence. Verbal violence was the most common form, followed by physical violence, while sexual violence was reported rarely. Physical violence included actions such as hitting, pushing, pulling hair, and pinching. Verbal violence involved insults, threats, cursing, accusation, name- calling, and shouting. Nurse 7 stated,

"We encountered physical, verbal, and sexual violence from patients and their relatives in the ED. Many people who come to the ED refuse to wait, even when the condition of their patient is not urgent. They insist on receiving treatment before other patients with serious conditions. This often end with shouting, and in many cases, escalates into beatings and aggression. Instance of sexual violence occurs rarely, but I have faced sexual harassment from male patients several times. On one occasion, a male patient attempted to harass a female nurse."

**Theme 2: Negative consequences of WPV**

Participants highlighted the adverse effects of workplace violence (WPV) on both their physical and mental well-being. WPV resulted in emotional distress, characterized by feelings of fear, irritability, anger, restlessness, sleep disturbances, frustration, and a diminished interest in work. Further consequences included exhaustion, absenteeism, injuries, and even fatalities. Nurse 16 expressed,

"The violent incident with the patient had a lasting impact on me, both psychologically and physically. At one point, I suffered from rib fractures that had not fully healed. When nurses encounter WPV, they often feel a loss of value, appreciation, or respect from the community. Over time, I found that WPV made me irritable, restless, and angry. Undoubtedly, the quality of my work has been dramatically affected."

Participants detailed the profound physical and emotional effects of workplace violence (WPV) on their family lives, significantly impacting their relationships and communication. Following incidents of WPV, nurses often found themselves easily agitated or angry, which had a dramatic influence on both their personal well-being and and their families.

Nurse 23 expressed, " Returning home after an exhausting shift and experiencing WPV affected my relationships with family members and relatives. I became more aggressive and nervous toward my children, losing my compassion for them, which seriously impacted their well-being.", which had serious repercussions for them."

Participants also highlighted the detrimental effects of WPV on healthcare institutions. Many nurses reported an increase in medical errors, the creation of unsafe environments for both staff and patients, and damage to resources. Additionally, WPV contributed to nurse burnout, intentions to leave their positions, and decreased job satisfaction, all of which diminished productivity and compromised the quality of patient care.

Nurse 10 remarked, "Violent acts by patients and their families can lead to destruction to resources and medical equipment in the ED. The ED became unsafe, and the hospital's reputation was jeopardized due to aggressive patient behaviors. One particularly unforgettable incident of WPV involved an addicted patient When the physician refused to prescribe him Pethidine. The patient pulled out a gun, threatened us, shouted, and damaged tools and equipment."

### Theme 3: Nurse's feelings their experience of WPV

Participants expressed the emotional turmoil they experienced following incidents of WPV. These incidents led to a range of profound feelings, including sadness, anger, nervousness, frustration, upset, injustice, persecution, enslavement, and humiliation. Nurses reported that WPV left them devastated, significantly affecting their overall lives.

Nurse 23 remarked, "WPV resulted in deep-seated frustration that lingers for a long time. Additionally, the experience filled me with anger, nervousness, and upset, ultimately leading to fatigue. I was left feeling humiliated after the incident, which caused a loss of motivation and creativity in my work."

In the long term, many nurses found themselves grappling with depression and emotional devastation. They reported a diminishing motivation and desire to work, accompanied by a persistent fear for their safety due to the ever-present threat of WPV. Nurses expressed their frustration stemming from the lack of appreciation for their efforts by patients and their families. Some even considered leaving their positions due to the overwhelming burden of WPV.

Nurse 14 noted, "The emergency department has become unsafe and unsecured. I feel that at any moment during my day, I could be subjected to violence from patients or their relatives. I have seriously thought about quitting my job, changing careers, or moving away from this violent environment."

### Theme 4: Coping strategies for dealing with WPV

Participants described their coping mechanisms after the incidents of WPV. These strategies included feigning ignorance about the situation, maintaining silence. and refraining from responding to patients to avoid further issues. Some nurses reported experiencing episodes of crying and shouting after WPV incidents. while others opted to withdraw from violent situations or even from the entire department. Nurse 5 said,

"In early stage my career, I felt outraged and anxious after WPV incidents. However, over time, and with the increasing frequency of these occurrences, I began to understand the nature of patients and their families, as well as their diverse cultural backgrounds and fears. My responses evolved; I chose to remain silent and distance myself from the violence. Occasionally, I would seek support from my fellow nurses."

### Discussion

In this descriptive qualitative study, twenty-four nurses shared their experiences with WPV in the ED. Nurses in the ED are regarded as some of the most vulnerable healthcare professionals to WPV due to their frontline roles and direct interactions with patients and their families. This exposure subjects them to significant stress and the risk of encountering WPV. Despite this, many nurses reported having insufficient knowledge and training regarding security measures and safety protocols related to WPV [1]. This situation underscores the necessity for enhanced education and training for medical staff and nurses to prevent incidents and effectively address them when they occur.

The nurses involved in this study provided detailed accounts of their experiences following WPV incidents in the ED, describing these events as hazardous, harsh, and uncommon. Many expressed shock at the aggressive behavior exhibited by patients during medical care.

In a study conducted by Ferri et al., nurses expressed feelings of shock, deep emotional distress, and a sense of unfairness regarding their experiences, as well as for their profession and colleagues, following an incident of WPV [2]. Furthermore, study participants in Jakobsson, Axelsson, and Örmon study described that WPV as an unpleasant and

hazardous experience, particularly when they faced physical attacks from delirious patients. They also recounted feelings of disrespect and fear when confronted with verbal threats from patients or their relatives, especially when these individuals wielded sharp knives or tools unexpectedly [17]. This highlights the importance on providing emotional sand legal support after the incidence of WPV.

Nurses in the current study reported experiencing high rates of workplace violence (WPV), with incidents occurring daily. Lee et al. found that 92.9% of nurses had encountered WPV in the previous two years [6]. A study conducted among Jordanian nurses in rural public revealed that 49.1% and 55.5% of nurses experienced physical and verbal violence [7].

Nurses participating in the current study described facing forms of aggressive verbal and physical attacks at work, particularly during night shifts, where female nurses were often targeted. In Lee et al.'s study, the most common types of WPV reported were verbal abuse (91.2%) and physical abuse (75.2% [6]. Furthermore, Ebrahim and Issa (2018) found that the highest incidence of WPV among nurses occurred during the night shift, affecting 48.7% of those surveyed [18].

In the study conducted by Jakobsson et al., nurses vividly recounted their experiences with physical violence, characterizing it as a distressing reality that includes actions such as beating, kicking, slapping, pushing, hitting, biting, and throwing objects at health staff. These acts are directed at individuals or groups, highlighting a troubling trend in workplace safety [17]. The CDC (2021) further emphasized that verbal abuse and threats manifest as offensive language, humiliation, and aggressive shouting, all designed to intimidate and demean individuals in their work environment [4].

In the current study, nurses reported instances of sexual violence, although they indicated that such cases were infrequent. Jakobsson et al.'s findings, participants revealed that sexual violence often manifested with derogatory language and unwelcome physical contact, with female nurses being particularly vulnerable to inappropriate touching [17]. These insights underscore a critical need for heightened awareness and protection for healthcare professionals including nurses in their work settings.

Nurses in the current study reported alarmingly high instances of workplace violence (WPV), yet they rarely conveyed these incidents to their health institutions. This silence is further compounded by the reluctance to discuss sexual violence, deeply rooted in Jordan's conservative culture. Alharthy et al.'s study reveals that the primary reasons cited for not reporting WPV were the belief that the reporting process was pointless (56%) and the perception that these incidents were not significant enough to warrant attention (52%). Shockingly, only 10% of WPV incidents were officially reported to health institutions [19].

In a related context, participants in the study from Jordan acknowledged WPV occurrences, but a mere 15 cases (10.8%) were escalated to legal action and intervention [8]. This stark reality underscores the urgent need for a culture of reporting and accountability within healthcare systems, emphasizing the crucial role of hospital authorities in addressing these serious issues.

Nurses in this study vividly demonstrated the profound effects of workplace violence (WPV) on their physical and mental health, as well as on the overall functioning of healthcare institutions and the quality of patient care. These findings are strikingly similar to those of Mento et al., which highlighted the severe consequences of WPV, including detrimental impacts on nurses' physical health and psychological well-being, ultimately affecting their conduct both professionally and personally [20]. The research by Mento et al. and Ferri et al. reveals a troubling pattern: WPV leads to significant harm to healthcare workers' psychological and physical health. It fosters heightened levels of stress and anxiety, spurs feelings of anger and fear, and can plunge individuals into depression, hopelessness, social isolation, and burnout [2,20]. This evidence underscores the urgent need to address WPV in healthcare settings to protect the well-being of both nurses and their patients.

Nurses in the current study powerfully articulated the profound and devastating impact of WPV on their family lives, severely disrupting relationships and communication. Participants in Varblik et al.'s research emphasized that WPV not only undermined their working environment but also significantly diminished productivity, strained family relationships, and disrupted their entire lives [21].

Moreover, nurses in the current study revealed alarming consequences of WPV for healthcare institutions, jeopardizing safety, diminishing staff productivity, and increasing turnover rates. In alignment with findings from Rasool et al., these nurses underscored the detrimental effects of WPV on their work environment and available resources. The repercussions were significant: reduced employee morale, hindered work performance and engagement, and striking declines in productivity, coupled with an increase in absenteeism. Ultimately, WPV eroded health workers' motivation and severely impacted their work efficiency, highlighting an urgent need for intervention and support [22].

In Adrianssen et al.'s study, nurses emphasized that workplace violence (WPV) not only damages hospital properties but also creates perilous environments for both patients and healthcare professionals [8]. Similarly, participants in Jakobssen et al.'s research pointed out that WPV significantly undermines patient care, causing delays, distracting nurses, and increasing the likelihood of errors [17]. Furthermore, Spelten et al.'s study revealed the far-reaching consequences of WPV on healthcare institutions and society at large, including soaring economic costs, a deterioration of public trust in healthcare providers, and the troubling normalization of violent behavior [23]. These findings underscore the urgent need to address workplace violence in healthcare settings to protect both patients and caregivers.

Nurses in the current study revealed the profound emotional toll they experienced after a distressing incident that irrevocably altered their lives. Over time, this trauma drained their motivation and passion for their work. They expressed heightened fears for their personal safety, having faced threats in their own workplace. Furthermore, they felt deep frustration as their dedication went unrecognized by patients and their families. This culmination of challenges compelled many to seriously contemplate abandoning their nursing careers for good.

These findings are consistent with the groundbreaking research by Itzhaki et al., which demonstrated that workplace violence (WPV) can have a devastating impact on nurses' professional quality of life. Those who viewed their work environment as hostile due to WPV reported alarmingly low job satisfaction and significantly higher rates of burnout ($r = 0.501$, $p < 0.01$). The consequences of WPV manifested as emotional exhaustion and a cascade of negative emotions, including sadness, anger, fear, and anxiety. Nurses experienced escalating job stress and dissatisfaction, ultimately leading to overwhelming frustration and an inability to meet the daily demands of their roles [24].

Participants in Sachdeva et al.'s study reported several negative impacts of workplace violence (WPV), including decreased job satisfaction in 54% of cases, feelings of fear in 37%, and sleep problems in 29% [12]. Furthermore, Jobe's research highlights the severe consequences of WPV in healthcare environments, revealing that it fosters fear, anxiety, and burnout among staff. These detrimental effects not only jeopardize the well-being of healthcare workers but also threaten staff retention, hinder productivity, and compromise the quality of patient care [25]. Addressing WPV is not just a moral imperative; it is essential for ensuring a safe and effective healthcare system.

Nurses in the current study reported various coping mechanisms in response to incidents of WPV. Many opted to ignore the disturbing situation, remain silent, or avoid confronting aggressive patients altogether. Others expressed their distress through tears or shouts, while others opted to withdraw from the scene or even the entire department

In the research conducted by Vrablik et al., participants outlined effective coping strategies, such as utilizing relaxation techniques and creating emotional and physical distance from the traumatic events. They emphasized the critical importance of debriefing after WPV incidents, employing avoidant coping methods, and fostering an environment where affected nurses could openly share their feelings with colleagues who had faced similar experiences [21].

The study by Zhou, Marchand, and Guay (2017) identified powerful strategies for managing WPV. Key among these was the importance of seeking assistance from professionals skilled in handling WPV instances, as well as reaching out for support from psychiatric experts and trusted friends, family, or colleagues [26].

The impact of WPV on nurses in our study was profound and deeply unsettling. Over time, and with repeated exposure to such violence, many nurses began to normalize and rationalize these incidents, adopting this mindset as a way to cope with the ongoing challenges they faced. Jakobsson et al.'s findings echoed this sentiment, illustrating how nurses came to accept WPV as an unfortunate, yet familiar aspect of their roles—one that occurred with alarming frequency [17].

Likewise, nurses in Spelten et al.'s study recognized that they had come to resign themselves to WPV as an inescapable reality of their careers, one that they felt powerless to change. Moreover, they highlighted the need to address more pressing societal issues rather than solely fixating on the pervasive problem of WPV [23].

## Limitations

This qualitative study engaged 24 Jordanian nurses, unveiling critical insights into the challenges they face. However, the findings are non-representative, which constrains their broader applicability beyond the specific participants involved. Consequently, these findings primarily reflect the experiences of emergency nurses in Jordanian emergency departments (EDs) and may not extend to other cultural settings or nursing specialties. To deepen our understanding, a comprehensive ethnonursing study could provide valuable context by incorporating nurses from diverse backgrounds within Jordan and beyond.

Furthermore, the exclusivity of our study participants—emergency nurses from just two public hospitals—limits the findings' relevance to comparable environments, leaving out potential insights from private facilities or hospitals in different cities. Importantly, the study did not include other healthcare professionals, such as physicians and paramedics, whose perspectives could significantly enrich our understanding of WPV in EDs.

This omission highlights a crucial gap in the research. The study also revealed a stark reality: many hospital settings are ill-prepared to combat WPV effectively. This underscores the urgent need for emergency departments to be equipped with adequate resources and for nurses and all healthcare staff to receive targeted training in managing WPV incidents. Addressing these issues is not just beneficial—it is essential for ensuring the safety and well-being of healthcare workers and patients alike.

## Study implications

Victimized emergency nurses must be empowered to report incidents of workplace violence (WPV) and to candidly share their experiences. It is imperative that they work hand-in-hand with hospital administrators to create robust programs that effectively combat WPV in the emergency department (ED). Comprehensive training for emergency nurses is essential, equipping them with the skills to manage dangerous situations, safely restrain patients, call for assistance, and prioritize their own safety during WPV incidents.

Nurse managers play a crucial role in advocating for and supporting emergency nurses who have faced violence, helping to avert the severe consequences WPV can have on their health, productivity, and the quality of patient care. Furthermore, the introduction of a universal screening tool for WPV is vital for accurately assessing the prevalence of violence in healthcare settings. This data will be instrumental in strategically allocating hospital resources to mitigate WPV, including implementing safety measures and alarm systems within the ED.

Enforcement of strong anti-WPV legislation and policies is not just advisable; it is essential for safeguarding the health and well-being of emergency nurses and ensuring the overall safety of hospitals. In Jordan, for instance, laws are in place that unequivocally criminalize any form of WPV against healthcare workers, including nurses during their shifts. Additionally, police presence in hospitals serves as a critical deterrent against violent acts, reinforcing the commitment to a safe working environment for all healthcare professionals.

## Conclusion

Nurses faced various forms of WPV and reported high incidence rates; however, there was a reluctance to report these incidents to hospital authorities. The ongoing WPV had a significant impact on the physical and mental health of the nurses, as well as on the safety and resources of healthcare institutions. Over time, these nurses felt compelled to normalize the occurrences of WPV, accepting it as part of their job, which further discouraged them from reporting incidents. As a result, they experienced feelings of sadness, anger, frustration, and helplessness. This troubling situation led to an

increase in medical errors and a decline in the quality of patient care and safety. Additionally, many nurses indicated their intention to leave their positions in search of a safer work environment, which could exacerbate the nursing shortage in healthcare institutions.

Neglecting the feelings of emergency nurses and normalizing the phenomenon of WPV is a concerning issue that requires urgent attention. It is essential to provide nurses with the necessary institutional support and assistance following WPV incidents. Furthermore, nurses should be encouraged to report these incidents to the appropriate legal authorities. Immediate care and intervention should be made available to affected nurses, along with serious legal consequences for perpetrators. Implementing anti-WPV prevention strategies is crucial, as is ensuring an immediate response to uphold nurses' safety and reduce turnover, particularly in the conservative cultural context of Jordan.

## Supporting information

**S1 File. xxx.**
(DOCX)

**S2 File. xxx.**
(DOCX)

**S3 File. xxx.**
(DOCX)

## Acknowledgments

The authors acknowledge nurses working at Emergency department at the two hospitals who participated and shared their experiences in this study and the Deanship of Research at Jordan University of Science and Technology for official support of the study.

## Author contributions

**Conceptualization:** Ahlam Al-Natour, Lubna Abuziad.

**Data curation:** Ahlam Al-Natour, Lubna Abuziad.

**Formal analysis:** Ahlam Al-Natour, Lubna Abuziad.

**Funding acquisition:** Ahlam Al-Natour, Lubna Abuziad.

**Investigation:** Ahlam Al-Natour, Lubna Abuziad.

**Methodology:** Ahlam Al-Natour, Lubna Abuziad.

**Project administration:** Ahlam Al-Natour, Lubna Abuziad.

**Resources:** Ahlam Al-Natour, Lubna Abuziad.

**Software:** Ahlam Al-Natour, Lubna Abuziad.

**Supervision:** Ahlam Al-Natour, Lubna Abuziad.

**Validation:** Ahlam Al-Natour, Lubna Abuziad.

**Visualization:** Ahlam Al-Natour, Lubna Abuziad.

**Writing – original draft:** Ahlam Al-Natour, Lubna Abuziad.

**Writing – review & editing:** Ahlam Al-Natour, Lubna Abuziad.

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
