## [Decision Letter · Decision Letter 0]

4 Mar 2025

PONE-D-24-52681“Emergency Department is not Safe Anymore: Nurses Describing their Suffering”PLOS ONE

Dear Dr. Al- Natour,

Thank you for submitting your manuscript to PLOS ONE. After careful consideration, we feel that it has merit but does not fully meet PLOS ONE’s publication criteria as it currently stands. Therefore, we invite you to submit a revised version of the manuscript that addresses the points raised during the review process.

We look forward to receiving your revised manuscript.

Kind regards,

Ahmad H. Al-Nawafleh, Ph.D, MPA, CI, RN

Academic Editor

PLOS ONE

Additional Editor Comments:

Dear Authors,

I hope you will consider all the comments of the three reviewers.

Looking forward to seeing your reply shortly.

All the best

Reviewers' comments:

Reviewer's Responses to Questions

**Comments to the Author**

1. Is the manuscript technically sound, and do the data support the conclusions?

Reviewer #1: Yes

Reviewer #2: Partly

Reviewer #3: Yes

2. Has the statistical analysis been performed appropriately and rigorously? 

Reviewer #1: I Don't Know

Reviewer #2: Yes

Reviewer #3: Yes

3. Have the authors made all data underlying the findings in their manuscript fully available?

Reviewer #1: Yes

Reviewer #2: Yes

Reviewer #3: Yes

4. Is the manuscript presented in an intelligible fashion and written in standard English?

Reviewer #1: Yes

Reviewer #2: No

Reviewer #3: Yes

5. Review Comments to the Author

Reviewer #1: I’m pleased to have this opportunity to review this interesting topic and manuscript. This qualitative descriptive study aimed to describe the experiences of workplace violence as perceived by Jordanian nurses working at the emergency department and describe workplace violence forms and nurses’ feelings and coping strategies toward workplace violence.

*Title and Abstract

The title and the abstract cover the main aspect of the work.

*Introduction

The introduction, overall, has a logical flow. Organizing in thematic way gives the introduction readability.

It would be better that the paragraph 5 (In recent years, WPV increased in Jordanian hospitals ….) to follow the paragraph 3 (Nurses who work in the Emergency Department (ED) are one of the…).

*Method

Nothing to mention in particular.

*Results

Table 2 needs some modification.

*Discussion

According to the study results, each result in each theme has been witnessed from previous studies in the discussion part.

Reviewer #2: overall it needs extensive English editing

for the introduction section many refrenced study results are included that must be in the discussion section, line 81 provide like recommendations for planning educational intervention

line 87 describes the present study result not an aim for the study

vague introduction

Methods lacks the limitations for the qualitative approach as researcher bias, also it lacks hospital characteristics that could affect WPV

what is the participants selection method used that gurantee "no bias"

The Colaizzi method need to be discussed in more details

Results: no clear explaination of how themes were derived

Discussion need to be more concise and focused to offer deeper interpretation as it focuses more on repitative comparisons without giving researcher's opinions why similarities or differences exist

Limitations: does not show how these limitations impacted the study

Conclusion: also too much repititions it need to be more concise

some references are too old

Reviewer #3: Reviewer comment and suggestions

Generally congratulation for the authors on writing this interesting topic concerning nurse’s impact at their work place. However several issues need improving.

•Adhere to journals guide line on organizing your work

•Work extensively to be clear grammar and typographical errors throughout the document

•Title does not specify what kind of danger or suffering nurses are experiencing

•Also the phrase emergency department is not safe anymore it seem like all emergency departments are unsafe which may not be entirely accurate

•Also on top of your title authors should write the study design

•On part of abstract method the authors need to improve to explain all procedure need

•Conclusion the authors should conclude according to result

•On part of Introduction the authors should revise its very narrow

•On part of result I comment to include the social demographic table it will be better.

•Also on part of question the authors should improve some of are similar idea and also the quotation the way reported the authors should use the scientific language used in research when reported.

•REFFERENCE Should be reversed are not clear in PLOS ONE guideline

•Red color noted line no 136-138 .also line 144-149

•Table are not seen

6. PLOS authors have the option to publish the peer review history of their article (what does this mean? ). If published, this will include your full peer review and any attached files.

**Do you want your identity to be public for this peer review?** For information about this choice, including consent withdrawal, please see our Privacy Policy .

Reviewer #1: **Yes: ** Evan Sabrah

Reviewer #2: No

Reviewer #3: **Yes: ** rehema abdallah

---

## [Author Response · Author response to Decision Letter 1]

22 Mar 2025

dear reviewers

thanks for your fruitful feedback . I went through every comment and did it as seen in the uploaded file

each reviewer comment were taken in consideration and changes will be seen all over the sent draft

thanks again

---

## [Editor Report · Decision Letter 1]

27 Mar 2025

“Emergency Department is not Safe Anymore: Nurses Describing their Suffering”

PONE-D-24-52681R1

Dear Dr. Al- Natour,

We’re pleased to inform you that your manuscript has been judged scientifically suitable for publication and will be formally accepted for publication once it meets all outstanding technical requirements.

Kind regards,

Ahmad H. Al-Nawafleh, Ph.D, MPA, CI, RN

Academic Editor

PLOS ONE
---

## [Editor Report · Acceptance letter]

PONE-D-24-52681R1

PLOS ONE

Dear Dr. Al- Natour,

I'm pleased to inform you that your manuscript has been deemed suitable for publication in PLOS ONE. Congratulations! Your manuscript is now being handed over to our production team.

Kind regards,

on behalf of

Dr. Ahmad H. Al-Nawafleh

Academic Editor

PLOS ONE